# Thermodynamic Properties and Equation of State for Solid and Liquid Aluminum

Nikolay V. Kozyrev *[ID] and Vladimir V. Gordeev [ID]

Laboratory for Physicochemical Fundamentals of Energetic Condensed Systems, Institute for Problems of Chemical and Energetic Technologies, Siberian Branch of the Russian Academy of Sciences (IPCET SB RAS), 659322 Biysk, Russia
* Correspondence: kozyrev@ipcet.ru

**Abstract:** High-temperature equations of state for solid and liquid aluminum were constructed herein using experimental data on thermodynamic properties, thermal expansion, compressibility, bulk modulus and sound velocity measurements, supplemented with phase diagram data (melting curve). The entire set of experimental data was optimized using the temperature-dependent Tait equation over a pressure range of up to 800 kbar and over a temperature range from 20 K to the melting point for solid aluminum and to 3800 K for liquid aluminum. The temperature dependence of thermodynamic and thermophysical parameters was described by an expanded Einstein model. The resultant equations of state describe well the totality of experimental data within measurement errors of individual variables.

**Keywords:** Tait equation of state; solid and liquid aluminum; melting curve; behavior modeling





## 1. Introduction

Thermodynamic and thermophysical properties of aluminum are the subject of numerous experimental and theoretical studies. Aluminum has a face-centered cubic (fcc) lattice under normal conditions and transits to a hexagonal close-packed (hcp) structure at a pressure of $2170 \pm 10$ kbar [1].

Aluminum and its alloys are commonly used in various fields of science and technology. A knowledge of the equation of state (EoS) is required to describe the aluminum behavior adequately and simulate Al-based composites over a wide pressure and temperature range. The currently available data on solid aluminum are limited either by isothermal compression at room temperature or by thermal expansion and elastic moduli measurements at normal pressure. Outside the thermodynamic parameters, only temperature-dependent density and sound velocity have been measured for liquid aluminum, with compressibility data lacking. The equations of state for solid and liquid phases suggested by Baker [2] using empirical power-law equations are not quite accurate in describing the thermal expansion and compressibility of both phases over a wide range of varying pressures and temperatures. That being the case, the way how the equation of state for liquid aluminum had been built was not reported. In this regard, it is worth noting that the existing experimental data for liquid aluminum are not enough for constructing a correct equation of state. However, using a pressure dependence of the melting point allows for an additional limitation on the liquid phase properties, as the chemical potentials of the solid and liquid phases at the point of melting are equal. In that case, an equation of state for the liquid phase can be derived by simultaneously co-processing all the thermodynamic and thermophysical experimental data for both phases. Such an approach offers extra merits. First, it permits thermodynamic and thermophysical data to be mutually agreed for each phase isolatedly and for both phases at a time. Second, it can soundly expand the equation of state for the solid phase towards higher pressures and temperatures, at least through to

the highest measured melting point in the phase diagram. Third, it can predict the liquid phase compressibility up to a maximum measured pressure in the melting curve.

It should be noted that finding equations of state by simultaneously co-processing the totality of thermodynamic and thermophysical data for a few phases with a melting curve involved is a challenging task. For instance, the study [3] failed to derive a realistic melting curve for aluminum based on the equality of chemical potentials of the distinct phases. The melting curve calculated in the study [2] at pressures of up to 350 kbar reproduces the experimental data quite inaccurately.

In our previous study [4], the aforesaid approach was employed to find equations of state for solid and liquid lead; however, the pressure range examined was confined to a pressure of 130 kbar (the existence domain of the fcc lead).

Therefore, it is of interest to explore if the said approach can be used for greater pressures. To this end, we chose aluminum whose properties have widely been studied in detail and melting curve measured at pressures up to ~800 kbar [5].

## 2. Physicochemical Model

Given the recommendations of Chase et al. [6], the thermodynamic variables were optimized herein by using an expanded Einstein model with a multiple function similar to that reported by Voronin et al. [7]. As stated by Jacobson and Stoupin [8], experimental data for a wide array of materials are fitted well by such a functional form with a varied number of terms.

### 2.1. Thermodynamic Functions

A trinomial Einstein equation with a correction power extra to take account of anharmonic effects was used herein to describe thermodynamic properties of solid aluminum in standard state. The thermodynamic functions of solid aluminum at zero pressure were adopted in the form:

$$H_T - H_0 = \sum_{i=1}^{3} \frac{Y_i \theta_i}{\exp(\theta_i/T_i) - 1} + hT^m \tag{1}$$

$$C_P = \frac{dH_T}{dT} = \sum_{i=1}^{3} \left(\frac{\theta_i}{T}\right)^2 \frac{Y_i \exp(\theta_i/T)}{[\exp(\theta_i/T) - 1]^2} + mhAT^{m-1} \tag{2}$$

$$S = \Delta S_0 + \sum_{i=1}^{3} Y_i \left\{ \frac{\theta_i}{T} \frac{\exp(\theta_i/T)}{\exp(\theta_i/T) - 1} - \ln[\exp(\theta_i/T) - 1] \right\} + \frac{m}{m-1} hT^{m-1} \tag{3}$$

where $T$ is the absolute temperature, $H$ is the enthalpy, $C_P$ is the isobaric heat capacity, $S$ is the entropy, $Y_i$, $\theta_i$, $h$ and $m$ are the constants, and $\Delta S_0$ is the integration constant.

The heat capacity of liquid aluminum in accord with different literature sources is invariable over a wide temperature range, starting from the melting point [9–14]. Therefore, the thermodynamic functions of liquid aluminum were adopted in the following form:

$$H_T - H_0 = aT + b \tag{4}$$

$$C_P = a \tag{5}$$

$$S = a \ln T + \Delta S_0 \tag{6}$$

where $a$, $b$ and $\Delta S_0$ are the constants.

The molar Gibbs free energy (chemical potential $\mu°$) is defined by the common relation:

$$G_m \equiv \mu° = [H_T - H_0] - TS \tag{7}$$

### 2.2. Molar Volume

The Tait equation was employed to describe the pressure-dependent molar volume of solid and liquid aluminum [15,16]. The density data for solid and liquid substances are

fitted well by this equation at pressures up to a few GPa [16]. The present study adopted the following high-temperature form of the Tait equation:

$$P = \frac{B_T}{n_0 + 1}\left\{\exp\left[(n_0 + 1)\left(1 - \frac{V}{V_T}\right)\right] - 1\right\} \tag{8}$$

where $P$ is the pressure, $V$ is the volume, $V_T$ and $B_T$ are the molar volume and the bulk modulus at zero pressure and temperature $T$, respectively, and $n_0$ is the pressure derivative of the bulk modulus. The temperature effect on molar volume $V$ was described via temperature-dependences $V_T$ and $B_T$.

The temperature-dependent molar volume of solid and liquid aluminum was described by different equations. To describe the thermal expansion of solid aluminum over a wide temperature range, a dependence similar to that for enthalpy (Equation (1)) was used:

$$\ln\left(\frac{V_T^S}{V_0^S}\right) = \sum_{i=1}^{3}\frac{X_i\Theta_i}{\exp(\Theta_i/T_i) - 1} + gT^k \tag{9}$$

where $V_T{}^S$ and $V_0{}^S$ are the molar volumes at zero pressure and temperatures $T$ and $T = 0$, respectively, and $X_i$, $\Theta_i$, $g$ and $k$ are the constants.

An inverse cubic function was employed to describe the temperature-dependent molar volume of liquid aluminum:

$$V_T^L = \frac{V_0^L}{1 + A_1 T + A_2 T^2 + A_3 T^3} \tag{10}$$

where $V_0{}^L$ is the "hypothetical" molar volume of liquid aluminum at zero temperature and zero pressure, and $A_1$, $A_2$ and $A_3$ are the constants.

Here, the melting point of aluminum at ambient pressure was assumed equal to 933.473 K [17].

## 2.3. Isothermal Bulk Modulus

The temperature-dependent isothermal bulk modulus of both solid and liquid aluminum was described by a function suggested by Deffrennes [18] for the estimation of isothermal compressibility:

$$K_T = K_0 + C\sum_i \frac{a_i}{\exp\left(\frac{\theta_i}{T}\right) - 1} \tag{11}$$

where $K_T$ and $K_0$ are the isothermal compressibility coefficients at temperatures $T$ and 0 K, respectively; $C$ and $\alpha_i$ are the constants contingent on the substance type. This equation was inverted to acquire the bulk modulus, and the analysis showed two terms in summation to be enough for an adequate description of experimental data. The expression in its final form is written as:

$$B_T = \frac{B_0}{1 + \sum\limits_{i=1}^{2}\frac{s_i}{\exp(\omega_i/T) - 1}} \tag{12}$$

where $B_T$ and $B_0$ are the bulk moduli at temperatures $T$ and 0 K, respectively; $w_i$ and $s_i$ are the constants. The advantage of this equation over the inverse quadratic dependence normally used [19] is that it adequately describes the measured data at low temperatures. Besides, this equation guarantees the modulus to be non-negative at very high temperatures.

*2.4. Melting Curve*

The molar Gibbs free energies (chemical potential) for solid ($\mu^S$) and liquid ($\mu^L$) phases in the melting plot must be equal [20]:

$$\mu^S = \mu^L \tag{13}$$

The isothermal variation in the chemical potential as the pressure changes is defined by the adopted equation of state and can be found via the relation [20]:

$$\mu - \mu^\circ = \int_{P^\circ}^{P} V dP \tag{14}$$

where $\mu$ and $\mu^\circ$ are the chemical potentials at pressure $P$ and reference pressure $P^\circ$, respectively. That variation is found by integration of Equation (8). The melting point at each pressure is estimated by simultaneous solution of Equations (7), (13) and (14) for solid and liquid phases.

## 3. Selected Experimental Data

*3.1. Thermodynamic Properties*

The thermodynamic properties of solid and liquid aluminum were overviewed in several reference books and review papers (Table 1).

**Table 1.** Data on thermodynamic properties of solid and liquid aluminum ($C_P$, J·mol$^{-1}$·K$^{-1}$) taken from reference books and review papers.

| $\Delta T$, K | Solid | | Liquid | Refs |
|---|---|---|---|---|
| | $C_P$ (298.15) | $C_P$ ($T_m$) | $C_P$ ($T_m$) | |
| 0.1–1700 | 24.339 | 33.867 | 31.756 | [9] (Buyco 1970) |
| 100–4500 | 24.354 | 33.881 | 31.750 | [10] (Glushko 1981) |
| 0–933.45 | 24.209 | 32.959 | – | [21] (Ditmars 1985) |
| 0.1–2800 | 24.225 | 33.107 | 31.757 | [11] (Desai 1987) |
| 298.15–2791 | 24.296 | 33.994 | 31.748 | [12] (Barin 1995) |
| 0–3000 | 24.209 | 32.959 | 31.751 | [13] (Chase 1998) |
| 298.15–1273 | 24.418 | 31.838 | 31.838 | [14] (Mills 2002) |

The heat capacity values at a temperature up to 300 K are well consistent with each other within 1%. The highest data scatter is observed at the melting temperature when the deviation relative to the data reported in [13] attains ±3.4%. It is worth noting that the reference book data [10] at temperatures above 300 K are based on the earlier work [9]. The thermodynamic functions of aluminum quoted in the studies [13,21] are almost coincident and on par with those reported in [11] within 0.5%.

According to all of the studies, the heat capacity of liquid aluminum at temperatures higher than the melting one is invariable and corresponds to the values listed in Table 1. The data are in good agreement with each other, except for the study reported in [14]. The heat capacity mean for liquid aluminum is 31.752 ± 0.004 J/(mol·K) according to the quoted studies (with disregard of [14]). Here, for liquid aluminum, we used the data from [13,21].

*3.2. Thermodynamic Properties of Solid Aluminum*

The experimental data on thermal expansion, isothermal compressibility and adiabatic bulk modulus were employed herein to construct an equation of state for solid aluminum. The isothermal bulk modulus included in the equation of state is defined by the common relation [22]:

$$B_T = \left( \frac{1}{B_S} + \frac{TV\alpha^2}{C_P} \right) \tag{15}$$

where $B_S$ is the adiabatic bulk modulus and $\alpha$ is the volumetric thermal expansion coefficient.

### 3.2.1. Molar Volume

Given that some studies report relative values of sample length or size rather than the absolute ones, it is required that a reference value be set for the molar volume of aluminum at standard temperature. Here, the molar volume of $9.996 \pm 0.001$ cm$^3$/mol (2.6992 g/cm$^3$ density) at 293.15 K, as calculated by Ablaster [23] who critically evaluated 46 experimental measurements from different literature sources, was taken as the reference value. The recalculation using the same thermal expansion coefficient as in [23] furnished a molar volume of $9.999 \pm 0.001$ cm$^3$/mol at standard temperature (298.15 K).

### 3.2.2. Thermal Expansion

An exhaustive literature overview and data treatment are provided in the studies reported in [23–25], wherein nearly all the references (more than 70) pertaining to the thermal expansion of solid aluminum can be found. The data on the thermal expansion coefficient given in those studies considerably differ between each other. The difference comes up to 6.5% in the low-temperature domain (50 K) and then declines to 1.2% at room temperature, and further rises again to reach 5.1% at 900 K. It should be noted herewith that the temperature-dependent molar volume data agree very well. The maximum difference is not in excess of 0.05% over the entire temperature range between 0 K and the melting point. Therefore, solely the molar volume data [23] were used herein, with the thermal expansion coefficient data left out of the optimization.

### 3.2.3. Isothermal Compressibility

The isothermal compressibility of aluminum under static compression at room temperature has experimentally been explored at pressures up to 41.9 kbar [26], 45 kbar [27], 70 kbar [28], 120 kbar [29], 200 kbar [30], 493 kbar [31], 1530 kbar [32], 2200 kbar [33], 2220 kbar [34], 3330 kbar [1] and 3680 kbar [35]. A fairly great spread of the data from different authors bears mention. For instance, in line with Greene et al. [33] and Akahama et al. [1], the same compression ratio of $V/V_0 = 6.0$ is achieved at pressures that differ by more than 8%, even despite the fact that both of the studies employed the same pressure calibration against the equation of state for platinum [36].

A series of studies were focused on estimating the normal isotherm of aluminum by restoring the shock-wave data at pressures up to 800 kbar [37] and 4000 kbar [38].

All the abovelisted studies in the original form were used herein for calculations at pressures up to 1000 kbar, except for the study by Dewaele et al. [35], who calibrated the pressure in their own way. The data from that study were recalculated using a more precise pressure calibration [39].

### 3.2.4. Adiabatic Bulk Modulus

The adiabatic bulk modulus was estimated from the elastic moduli measured over a temperature range of 4–300 K [40,41], 293–925 K [42] and 83–298.15 K [43]. Some studies performed measurements at room temperature only [44–47] and at 80 K [48]. Ho and Ruoff [49] estimated the isothermal bulk modulus using the elastic moduli measured at between 77 and 300 K. The data from all the studies agree well with each other. The adiabatic bulk modulus mean at 300 K calculated from all the measurement data is $762.0 \pm 8.0$ kbar ($\pm 1.0\%$). Sutton [50], and Tallon and Wolfenden [51], performed measurements over a wide temperature range of 63–773 K and 273–913 K, respectively. However, the reported data considerably differ from those of the other authors and were disregarded in the calculation. Apart from the said studies, the results from pressure-dependent adiabatic bulk modulus measurements in a range of 2.6–41.9 kbar were also factored in when performing the optimization herein.

### 3.3. Thermophysical Properties of Liquid Aluminum

There are no directly measured data on isothermal compressibility and bulk modulus of liquid aluminum in the literature. Therefore, we used measurement results of thermal expansion and sound velocity. The adiabatic bulk modulus was calculated by the common relation [52]:

$$B_S \equiv -V \left( \frac{dP}{dV} \right)_S = \rho u_S^2 \tag{16}$$

where $\rho$ is the density and $u_s$ is the sound velocity. The isothermal bulk modulus included in the equation of state was estimated by Equation (15).

#### 3.3.1. Thermal Expansion

The studies on thermal expansion of liquid aluminum were reviewed in [53,54]. In both papers, a linear temperature-dependence of density was used. However, these review papers left aside most of the literature sources numbering more than 40. Therefore, we performed a critical overview and selected the most reliable studies that are listed in Table 2.

**Table 2.** A list of selected studies on thermal expansion of liquid aluminum.

| $\Delta T$, K | Purity, % | Form [a] | $\Delta \rho$ [b], % | Refs |
|---|---|---|---|---|
| 933–1173 | 99.998 | P | – | [55] (Glazov 1958) |
| 1264–1733 | 99.99 | T, P, E | 0.5 | [56] (Goltsova 1965) |
| 933–1750 | 99.99 | P, E | ±1 | [57] (Ayushina 1968) |
| 933–1250 | 99.99 | P, E | 1.5 | [58] (Yatsenko 1972) |
| 933–1473 | 99.99 | E | ±0.5 | [59] (Bykova 1974) |
| 933–1340 | 99.999 | P, E | 0.2 | [60] (Drotning 1979) |
| 933–2070 | 99.999 | T, E | 0.2 | [61] (Makeev 1989) |
| 973–1173 | 99.99 | P, E | – | [62] (Smith 1999) |
| 1639–2360 | 99.99 | P, E | 1.5 | [63] (Sarou-Kanian 2003) |
| 933.6–1200 | – | P, E | – | [64] (Hairulin 2003) |
| 938–1113 | 99.999 | T, P | | [65] (Srirangam 2011) |
| 933–1673 | 99.999 | P, E | 1 | [66] (Schmitz 2012) |
| 933–1643 | 99.999 | P, E | 0.2 | [67] (Kurochkin 2013) |
| 933–1680 | 99.999 | E | ±3.8 | [68] (Leitner 2017) |
| 933–1823 | 99.999 | P, E | ±1 | [69] (Wessing 2017) |
| 933–1270 | – | P, E | 0.2 | [70] (Rusanov 2018) |
| 1356–1743 | 99.999 | P, E | 1 | [71] (Gancarz 2018) |

[a] Form of data: D—individual measurements; E—equation; P—plot; T—Table. [b] Reported error (%).

The average density of liquid aluminum at the melting temperature, calculated from all the measured data, was $2.375 \pm 0.005$ g/cm$^3$.

#### 3.3.2. Sound Velocity

The studies on sound velocity measurements in liquid aluminum were reviewed by Blairs [72]. Here, we employed the sound velocity measurement results reported in [73–80].

### 3.4. Melting Curve

The aluminum melting curve has experimentally been examined at pressures up to 14 kbar [81], 50 kbar [22], 100 kbar [82], 493 kbar [31] and 770 kbar [5]. Hanstrom [31] fitted experimental data by the Simons formula [83].

The aluminum melting curve was theoretically modelled by different techniques in a range of studies. In most studies, the estimated data were only represented graphically. References to the theoretical studies can be found elsewhere [84–87], where the melting curve was modelled at pressures not above 2000–5000 kbar, and the calculation results were tabulated [84] or fitted by the Simons formula [85–87].

## 4. Calculation Procedure

The error function representing a weighted root-mean-square deviation was adopted as an optimization criterion:

$$R = \sqrt{\frac{1}{N}\left[\sum_{i=1}^{N} w_i^2 \left(\frac{D_i^c - D_i^m}{D_i^m}\right)^2\right]} \tag{17}$$

where $N$ is the total number of experimental points; $D_i$ is the values of different parameters (enthalpy, heat capacity, molar volume, etc.); and $w_i$ is the weighting coefficients of these parameters. Indices $c$ and $m$ are the calculated and measured properties, respectively. The weighting coefficients were evaluated using relative measurement errors of different parameters. At each specified (experimental) pressure, a temperature at which the given chemical potentials (chemical potential divided by temperature) of solid and liquid phases differed at most by $10^{-8}$ J/(mol·K) was taken as the melting one.

The function was minimized via the Nelder–Mead simplex method for multidimensional minimization [88].

## 5. Results and Discussion

All the parameters resulted from the optimization are summarized in Table 3 for solid aluminum and in Table 4 for liquid aluminum. A comparison with the measured data is displayed in Figures 1–8.

**Table 3.** A summary of optimized variables of the EoS for solid aluminum.

| Equation | Parameter | Value |
|---|---|---|
| Thermodynamic functions (**1**)–(**3**) | $\Upsilon_1$, J·mol$^{-1}$·K$^{-1}$ | 0.4307400 |
| | $\Upsilon_2$, J·mol$^{-1}$·K$^{-1}$ | 11.46590 |
| | $\Upsilon_3$, J·mol$^{-1}$·K$^{-1}$ | 14.01224 |
| | $\theta_1$, K | 64.9599 |
| | $\theta_2$, K | 208.0659 |
| | $\theta_3$, K | 392.5907 |
| | $\Delta S_0$, J·mol$^{-1}$·K$^{-1}$ | 0.015206 |
| | $h$, J·mol$^{-1}$·K$^{-m}$ | $5.346947 \times 10^{-7}$ |
| | $m$ | 3.227389 |
| EoS (**8**) | $n_0$ | 4.69557 |
| Bulk modulus (**12**) | $B_0$, kbar | 795.69 |
| | $s_1$ | 0.10356 |
| | $s_2$ | 5.3621 |
| | $\omega_1$, K | 225.08 |
| | $\omega_2$, K | 3980.5 |
| Thermal expansion (**9**) | $V_0$, cm$^3$/mol | 9.87109 |
| | $X_1$ | $2.4646 \times 10^{-5}$ |
| | $X_2$ | $4.5046 \times 10^{-5}$ |
| | $X_3$ | $1.5699 \times 10^{-4}$ |
| | $\Theta_1$, K | 191.6603 |
| | $\Theta_2$, K | 368.4523 |
| | $\Theta_3$, K | 4244.7220 |
| | $g$, K$^{-k}$ | $2.4407 \times 10^{-8}$ |
| | $k$ | 1.840799 |

**Table 4.** A summary of optimized variables of the EoS for liquid aluminum.

| Equation | Parameter | Value |
|---|---|---|
| Thermodynamic functions (**4**)–(**6**) | $a$, J·mol$^{-1}$·K$^{-1}$ | 31.75 |
| | $b$, J·mol$^{-1}$·K$^{-1}$ | 3755.104 |
| | $\Delta S_0$, J·mol$^{-1}$·K$^{-1}$ | $-145.7490$ |
| EoS (**8**) | $n_0$ | 5.22253 |
| Bulk modulus (**12**) | $B_0$, kbar | 538.96 |
| | $s_1$ | 0.14097 |
| | $s_2$ | 2.5298 |
| | $\omega_1$, K | 676.33 |
| | $\omega_2$, K | 2668.4 |
| Thermal expansion (**10**) | $V_0{}^L$, cm$^3$/mol | 10.20834 |
| | $A_1$, K$^{-1}$ | $-1.0952 \times 10^{-4}$ |
| | $A_2$, K$^{-2}$ | $-1.0672 \times 10^{-10}$ |
| | $A_3$, K$^{-3}$ | $3.5504 \times 10^{-13}$ |

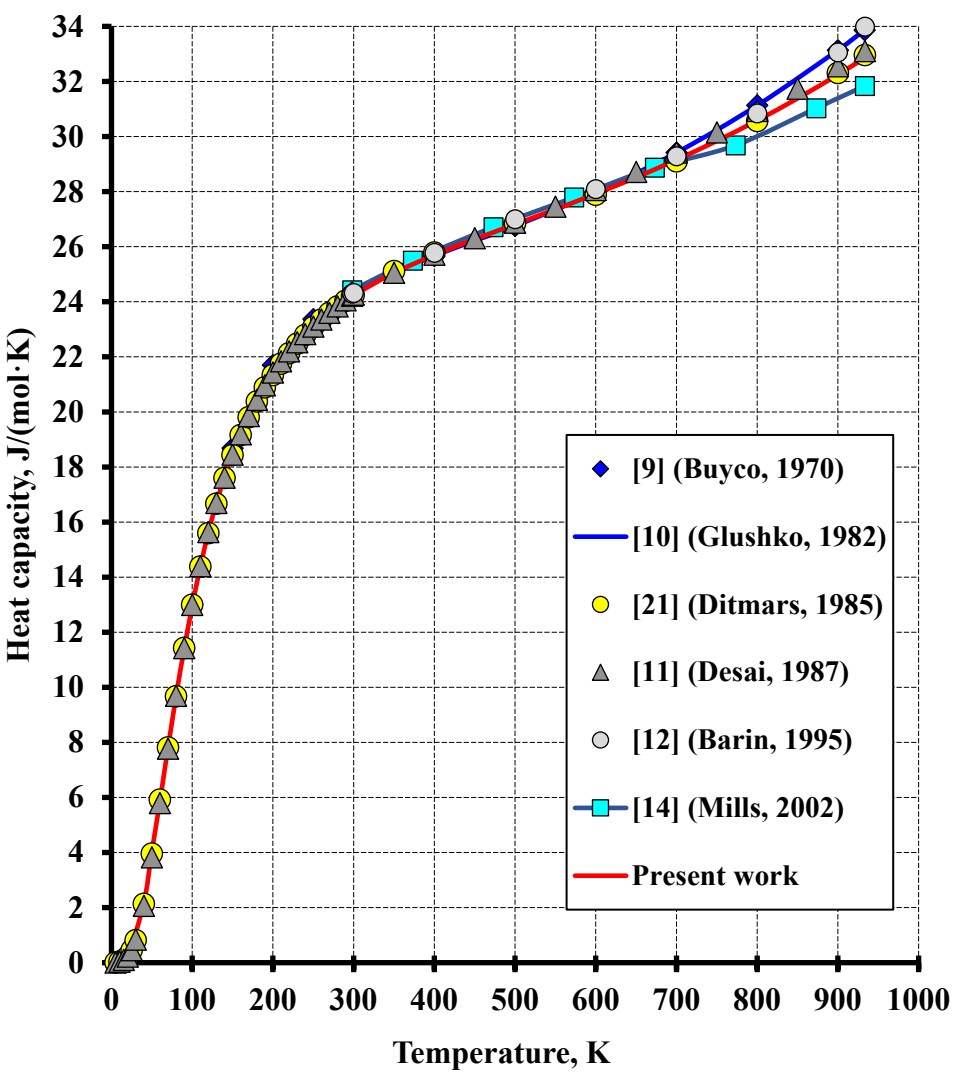

**Figure 1.** Isobaric heat capacity of solid aluminum.

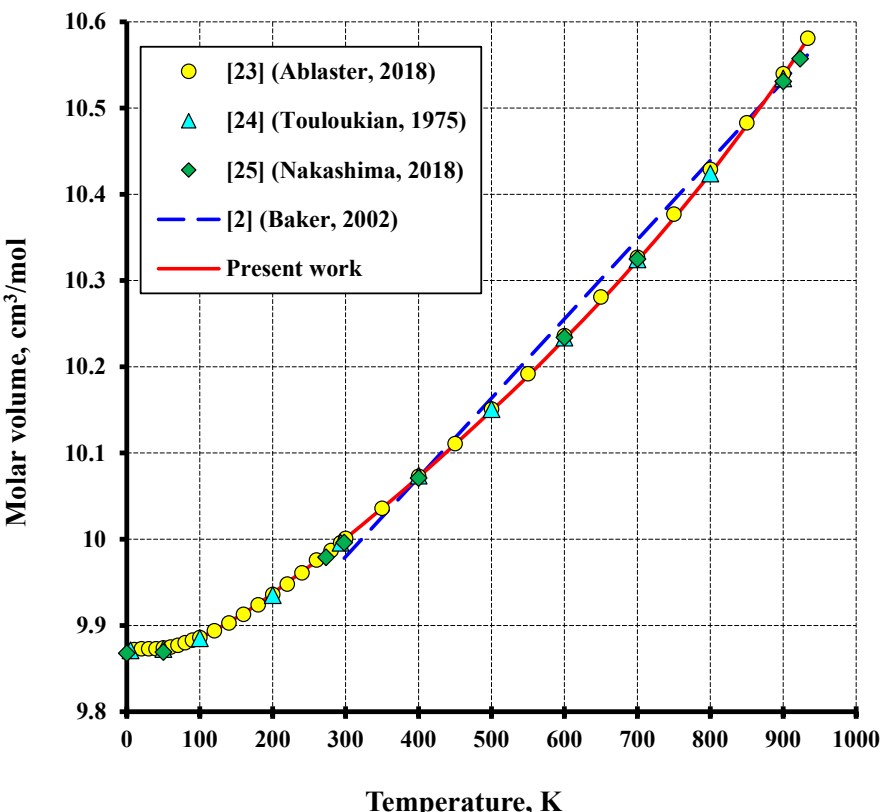

**Figure 2.** The molar volume of solid aluminum plotted versus temperature.

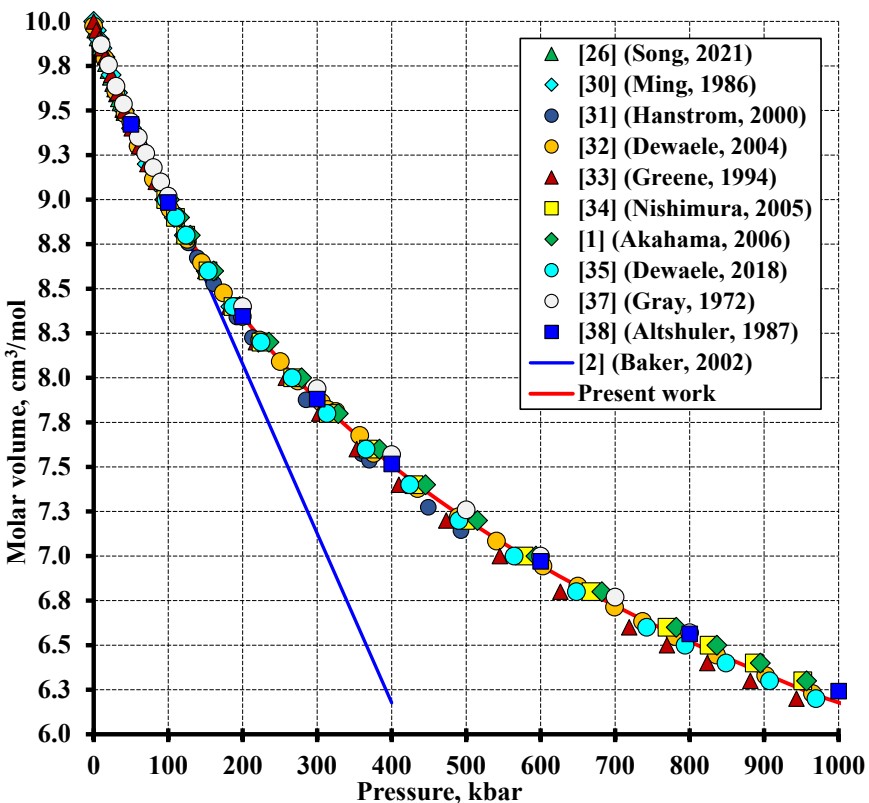

**Figure 3.** Volume compression of aluminum at 25 °C.

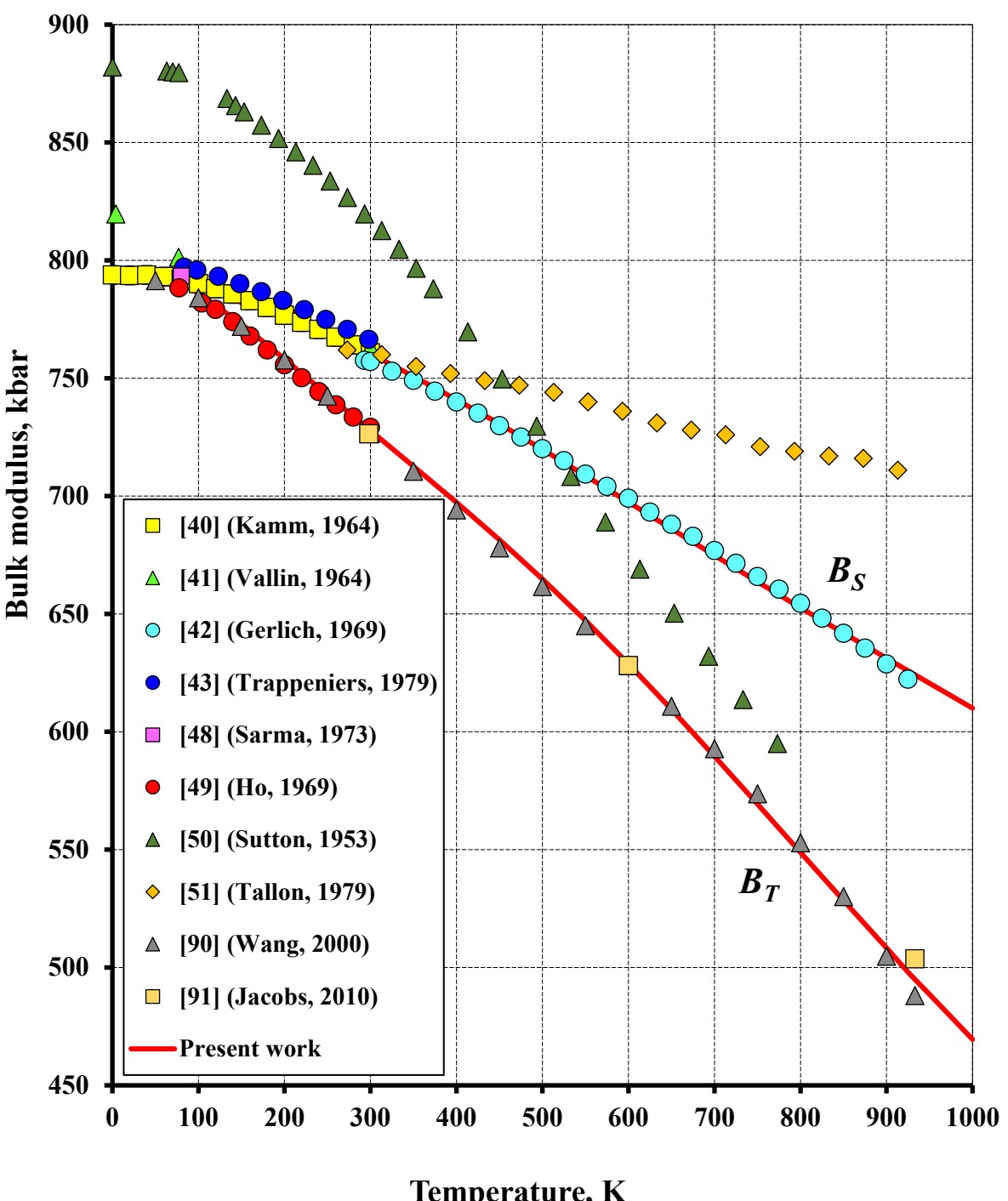

**Figure 4.** The adiabatic ($B_S$) and isothermal ($B_T$) bulk moduli of aluminum plotted against temperature.

Figure 1 compares the calculated heat capacity of solid aluminum with that from the other literature sources. The dependence obtained herein approximates well the data reported in [21]. The average root-mean-square (RMS) deviation between the calculated and measured data over a temperature range between 20 K and the melting point was 0.7%. The deviations were lower for the enthalpy: RMS = 0.25% for the entire temperature range.

Figure 2 depicts the molar volume of solid aluminum plotted against temperature. The experimental data are fitted well by the calculated dependence. The mean absolute deviation was 0.0017 cm$^3$/mol (RMS = 0.022%). The calculated molar volume at 298.15 K was 9.999 cm$^3$/mol and is not different from the reference value. At 0 K, the calculated molar volume of 9.871 cm$^3$/mol is distinct by 0.02% from that reported in [89].

Figure 3 illustrates the molar volume of aluminum plotted against pressures up to 1000 kbar. The experimental points totaled 113. The mean absolute deviation between the calculation and experiment over the entire pressure range was 0.018 cm$^3$/mol, RMS = 0.30%. Table 5 compares the Tait equation of state parameters obtained herein with those from the literature. The calculated isothermal bulk modulus at 298.15 K was 728.6 kbar and is on par with the literature data listed in Table 5.

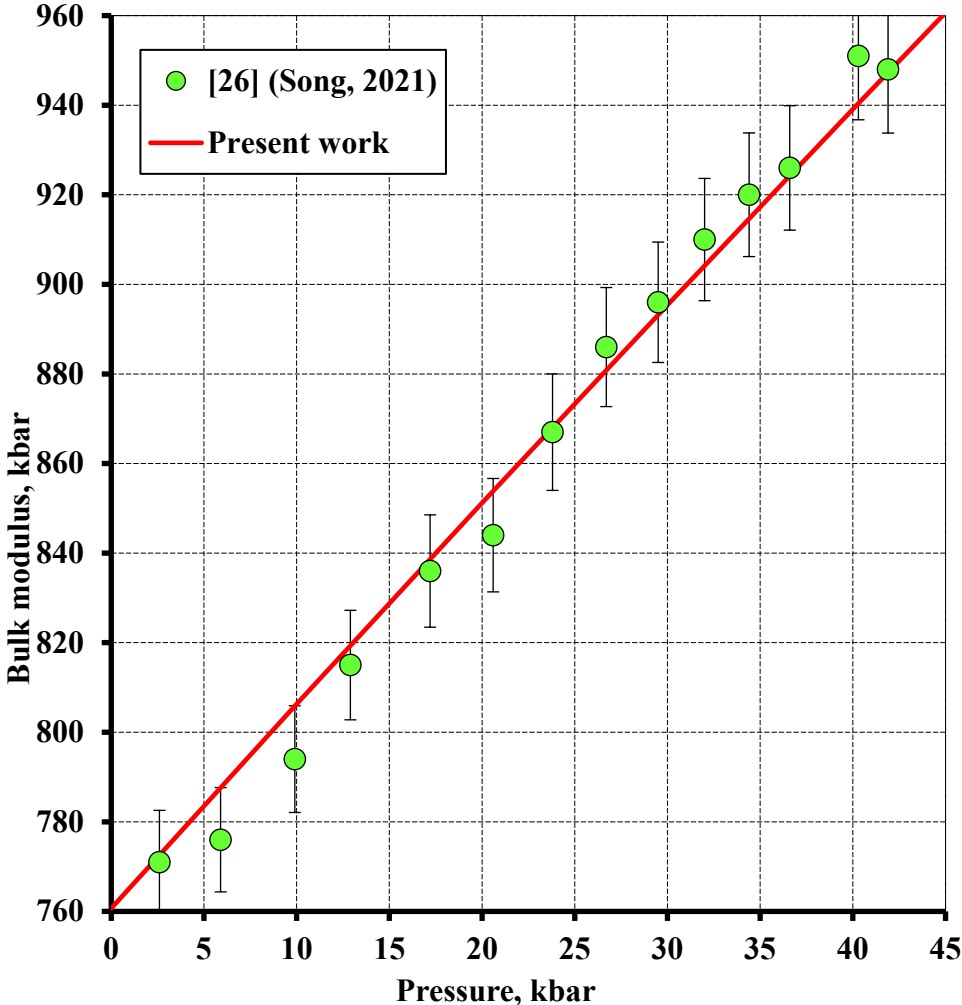

**Figure 5.** The adiabatic bulk modulus of aluminum plotted against pressure.

**Table 5.** Equation of state parameters for aluminum.

| $V_0$ [1], cm$^3$/mol | $B_0$ [1], kbar | $n_0$ | EoS [2] | Refs |
|---|---|---|---|---|
| 9.993 | 778.97 | 4.26 | Mur | [27] (Vaidya 1970) |
| – | 730 | 4.1 | Mur | [28] (Senoo 1976) |
| 9.998 | 727 (30) | 4.30 (8) | BM3 | [29] (Syassen 1978) |
| – | 717 (36) | 4.79 (37) | BM3 | [30] (Ming 1986) |
| 9.973 | 743 (11) | 4.47 (6) | Vinet | [32] (Dewaele 2004) |
| – | 727 | 4.14 | BM3 | [33] (Greene 1994) |
| 9.995 | 727 (20) | 4.446 (83) | BM3 | [34] (Nishimura 2005) |
| – | 760 (20) | 4.6 (7) | Vinet | [1] (Akahama 2006) |
| 9.98 | 730 | 4.54 (2) | Vinet | [35] (Dewaele 2018) |
| 9.871 [3] | 795.69 [3] | 4.696 | Tait | Present work |

[1] At 298 K. [2] Mur: Murnaghan EoS; BM3: third-order Birch–Murnaghan EoS; Vinet: Rydberg–Vinet EoS. [3] At 0 K.

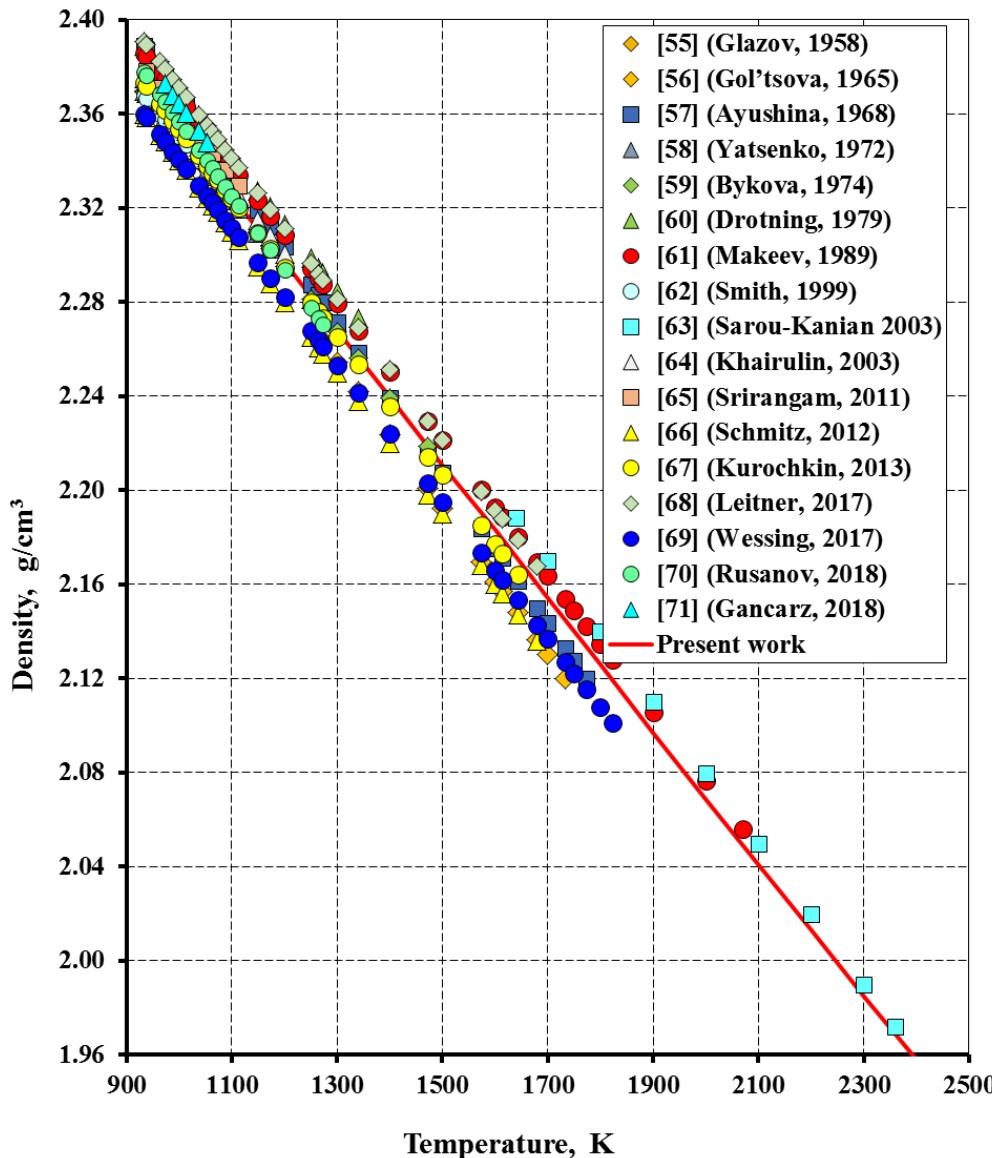

**Figure 6.** Density of liquid aluminum plotted versus temperature.

The bulk modulus of solid aluminum is plotted against temperature in Figure 4 and against pressure in Figure 5. The temperature dependence of adiabatic bulk modulus obtained herein reproduces the experimental data [40,42,43] within a measurement error. The calculated isothermal bulk modulus almost matches the data from [49] and is well consistent with the calculation results reported in [90,91]. The pressure dependence of adiabatic bulk modulus is in agreement with the experiment [26] within a measurement error (Figure 5).

The temperature-dependent density of liquid aluminum is depicted in Figure 6. The calculated density of 2.373 g/cm$^3$ for liquid aluminum at the melting temperature is different by 0.08% from the experimental average.

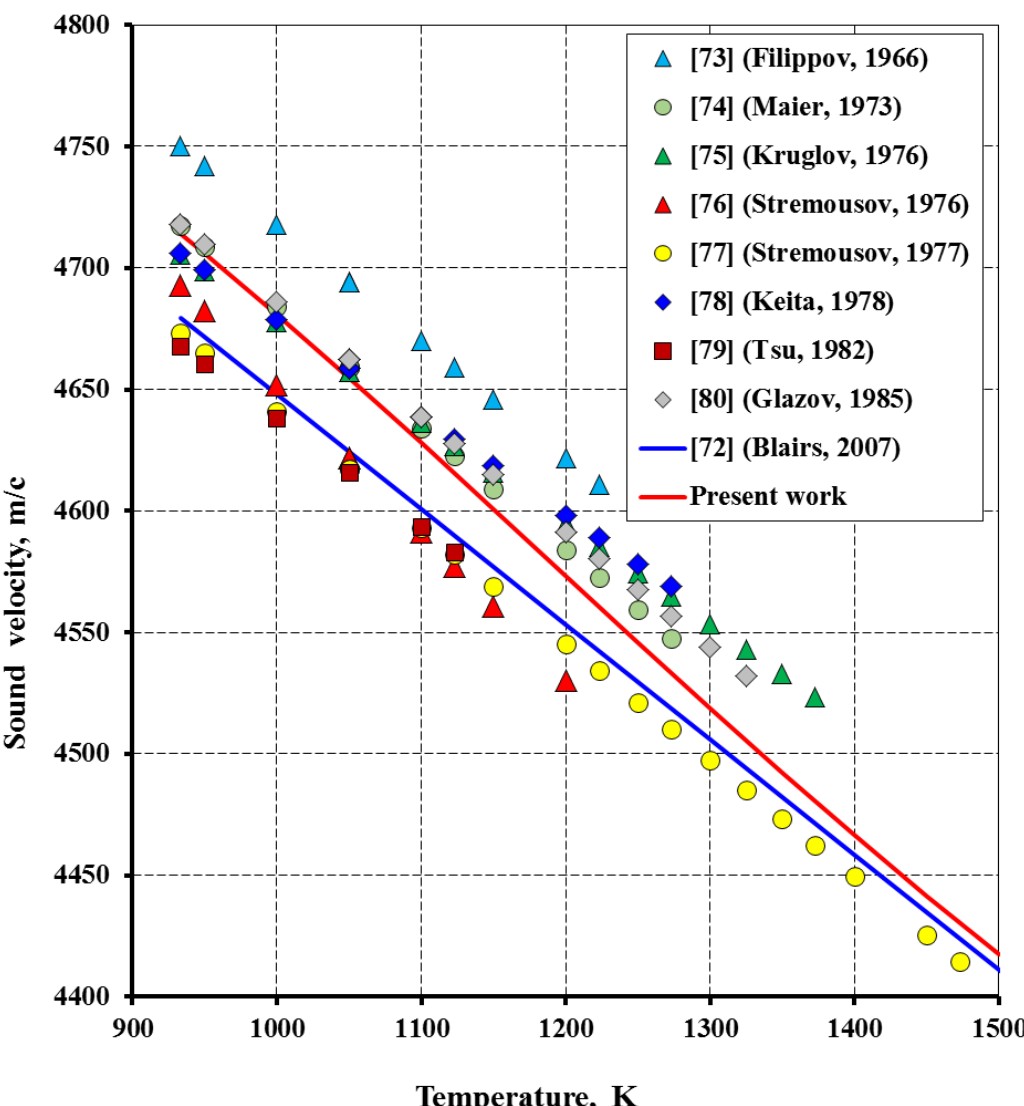

**Figure 7.** Sound velocity in liquid aluminum.

The sound velocity in liquid aluminum as a function of temperature is shown in Figure 7. The deviation between the calculated sound velocities and the data reported in [72] is 0.7% at the melting temperature and diminishes to 0.15% at 1500 K.

The aluminum melting curve calculated in the present study is compared with the experimental and theoretical data in Figure 8. The calculated curve reproduces the measured data within an error and is well consistent with the theoretical modelling results reported in [85,86]. The calculated curves lie below the experimental data. The calculated temperature at a pressure of 0.1 MPa is 933.470 K and is almost coincident with the measured value of 933.773 K [17].

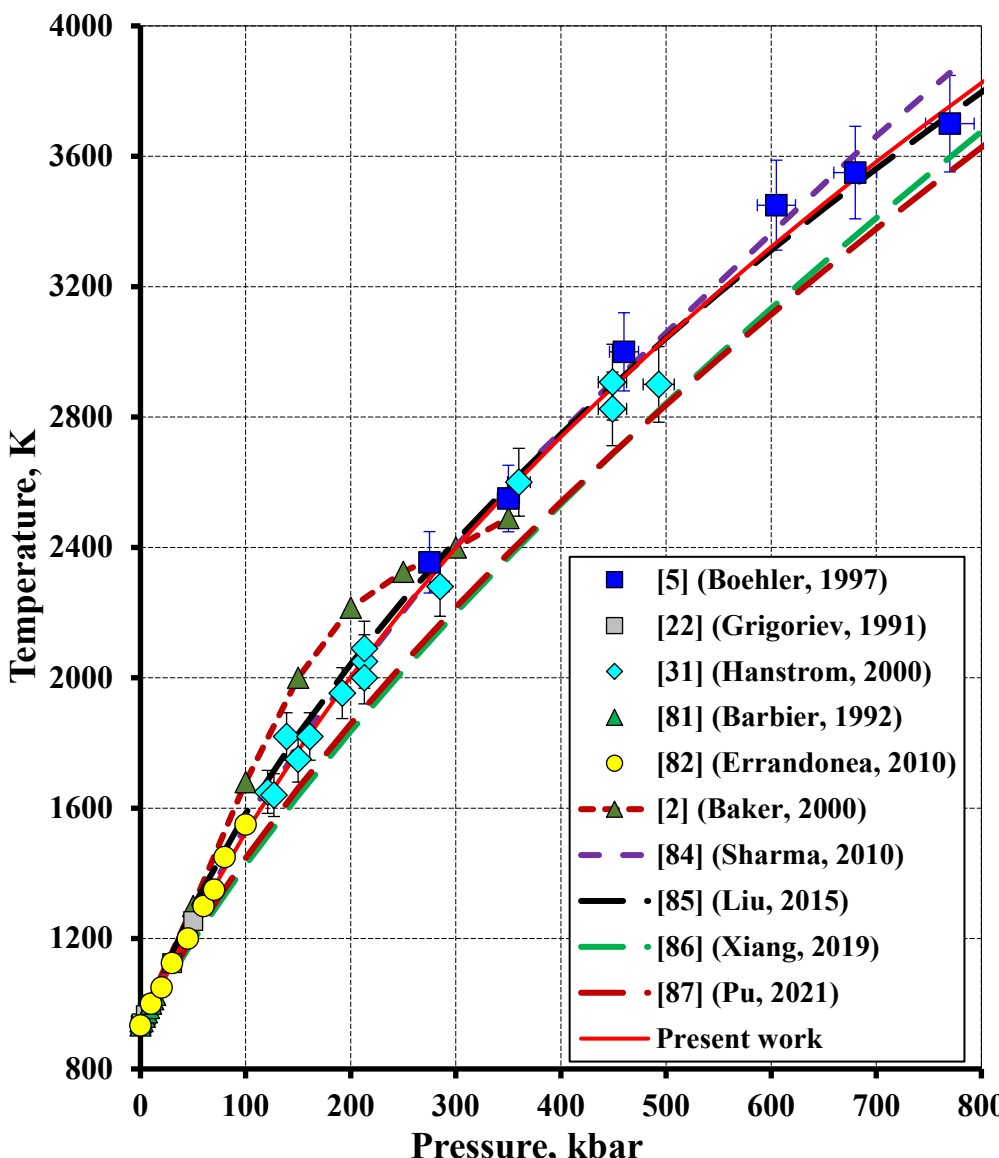

**Figure 8.** The calculated melting curve of aluminum as compared to experimental [5,22,31,81,82], calculated [2] and theoretical data [84–87].

## 6. Conclusions

The findings from this study demonstrate that the model used herein enables the description of experimental data for solid and liquid aluminum over a wide pressure and temperature range within an experimental error of measurement. That said, the thermodynamic and thermophysical variables for solid aluminum have been mutually agreed with each other by using common thermodynamic relations over a pressure range up to 800 kbar and over a temperature range from 10 K to the melting point. An equation of state for liquid aluminum has been constructed that can be used to predict the aluminum behavior at temperatures up to 3800 K in the specified pressure range. Moreover, all the thermodynamic and thermophysical parameters of solid and liquid aluminum have been harmonized with each other through the use of the melting curve in the co-optimization. The prediction accuracy can be enhanced and the applicability limits of the equations of state constructed against pressure and temperature can be broadened considerably by refining and expanding the melt curve data.

**Author Contributions:** Conceptualization, investigation, software, writing—original draft preparation, writing—review and editing, supervision: N.V.K.; investigation: V.V.G. All authors have read and agreed to the published version of the manuscript.

**Funding:** This research was supported by the Ministry of Science and Higher Education of the Russian Federation (under agreement No. 075-15-2020-803 with the Zelinsky Institute of Organic Chemistry RAS).

**Institutional Review Board Statement:** Not applicable.

**Informed Consent Statement:** Not applicable.

**Data Availability Statement:** The data that support the findings of this study are available from the corresponding author upon reasonable request.

**Conflicts of Interest:** The authors declare no conflict of interest.

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
