# Peer review of "Thermodynamic Properties and Equation of State for Solid and Liquid Aluminum"

_metals, doi:10.3390/met12081346_

Round 1

Reviewer 1 Report

This work presents high-temperature equations of state for solid and liquid aluminum using experimental data on thermodynamic properties, thermal expansion, compressibility, bulk modulus and so on. The resultant equations of state describe well the totality of experimental data within measurement errors of individual variables. The goal of this paper is very clearly stated, and the experiments and subsequent analysis partially support the intended goal of the paper. I recommend this manuscript for publication after minor revision.

1. There were some grammar mistakes and spelling mistakes. For instance, Line 228, Page 6, it should be “us but not us”. Please check the whole manuscript.

Author Response

The response to the reviewer’s comments has been uploaded as a separate pdf file.

Reviewer 2 Report

In this paper by Kozyrev and Gordeev, the authors are applying a temperature dependent Tait equation of state and an expanded Einstein model to a various set of experimental data, to model the equations of state (EOS) of solid and liquid aluminium (Al).

According to the authors, the present method has already been successfully used by them on lead but, in a much smaller pressure (P)-temperature (T) range. Therefore, they want to show its applicability at higher P-T, as well as its improved precision with respect to the model reported by Baker [ref 2].

Although the authors are applying their method to several experimental data obtaining convincing graphs, the actual description of both the methodology and the obtained results are not discussed in major details. Furthermore, there is no direct comparison between the results obtained with the present model and the one reported in [2].

Although the present work does not provide any additional information on the phase diagram and EoS of Al, the ability of the present model to describe the numerous results reported in literature is undeniable. Therefore, although the advances reported in this work are only incremental, I think it deserves to be published in Metals after some minor revisions.

In particular:

i)                    In the abstract you are talking about sound velocity, not speed velocity.

ii)                   In the introduction, reference [1] should refere to the work of Akahame et al. PRL 2006, 96, 045505, not the one from McMahon (that can be kept as additional one)

iii)                 Although the authors are referring to the fact the model used by Baker [2] based on empirical power-law is not accurate, they are not showing any direct comparison of the results obtained with the two different models with respect to the experimental data. Therefore, their statement cannot be verified by the community.

iv)                  End of line 60. A reference is needed.

v)                   Lines 77-78: “The heat capacity of liquid aluminium in accord with different literature sources….” References are needed for the sources.

vi)                 Lines 97. The actual temperature-dependence of Vt and Bt is missing!

vii)               Line 110. “The melting point of aluminium..” at ambient P must be added.

viii)         Table 5. Why is your Al, way more incompressible that the one obtained in all the other studies?

ix)     The obtained results are often compared only with a selected set of experimental data, and do not provide an actual overview with respect to all the experimental data and the physical consequences.

Author Response

(The authors gave the same response as above.)

Author Response

(The authors gave the same response as above.)

Round 2

Reviewer 3 Report

I have no further comments and suggestions and recommend publication in the present form.